# Long-range transported bioaerosols captured in snow cover on Mount Tateyama, Japan: Impacts of Asian-dust events on airborne bacterial dynamics relating to ice-nucleation activities

Teruya Maki<sup>1</sup>, Shogo Furumoto<sup>1</sup>, Yuya Asahi<sup>1</sup>, Kevin C. Lee<sup>2</sup>, Koichi Watanabe<sup>3</sup>, Kazuma Aoki<sup>4</sup>, Masataka Murakami<sup>5</sup>, Takuya Tajiri<sup>5</sup>, Hiroshi Hasegawa<sup>1</sup>, Asami Mashio<sup>1</sup> and Yasunobu Iwasaka<sup>6</sup>

<sup>1</sup>College of Science and Engineering, Kanazawa University, Kakuma, Kanazawa, Ishikawa, 920-1192, Japan.
 <sup>2</sup>School of Science, Auckland University of Technology, Private Bag 92006, Auckland 1142, New Zealand.
 <sup>3</sup>Department of Environmental and Civil Engineering, Faculty of Engineering, Toyama Prefectural University, 5180 Kurokawa, Imizu, Toyama 939-0398, Japan.

<sup>4</sup>Department of Earth Sciences, Faculty of Science, University of Toyama 3190 Gofuku, Toyama 930-8555, Japan. <sup>5</sup>Cloud Physics Section, Forecast Research Department, Meteorological Research Institute, Nagamine 1-1, Tsukuba, Ibaraki 305-0052, Japan.

15

5

Correspondence to: Teruya Maki (makiteru@se.kanazawa-u.ac.jp)

Abstract. The westerly wind travelling at high altitudes over East Asia transports aerosols from the Asian deserts and urban areas to downwind areas such as Japan. These long-range transported aerosols include not only mineral particles, but also microbial particles (bioaerosols), that impact the ice-cloud formation processes as ice nuclei. However, the detailed relations

- of airborne bacterial dynamics to ice nucleation in high-elevation aerosols have not been investigated. Here, we used the aerosol particles captured in the snow cover at altitudes of 2,450 m on Mt. Tateyama to investigate sequential changes in the ice-nucleation activities and bacterial communities in aerosols and elucidate the relationships between the two processes. After stratification of the snow layers formed on the walls of a snow pit on Mt. Tateyama, snow samples, including aerosol particles, were collected from 70 layers at the lower (winter accumulation) and upper (spring accumulation) parts of the
- snow wall. The aerosols recorded in the lower parts mainly came from Siberia (Russia), North Asia, and the Sea of Japan, whereas those in the upper parts showed an increase in Asian dust particles originating from the desert regions and industrial coasts of Asia. The snow samples exhibited high levels of ice nucleation corresponding to the increase of Asian dust particles. Amplicon sequencing analysis using 16S rRNA genes revealed that the bacterial communities in the snow samples predominately included plant associated and marine bacteria (phyla Proteobacteria) during winter; whereas, during spring,
- when dust events arrived frequently, the majority were terrestrial bacteria of phyla Actinobacteria and Firmicutes. The relative abundances of Firmicutes (Bacilli) showed a significant positive relationship to the ice nucleation in snow samples. Presumably, Asian dust events change the airborne bacterial communities over Mt. Tateyama and carry terrestrial bacterial populations, which possibly induce ice-nucleation activities, thereby indirectly impacting upon climate changes.

<sup>&</sup>lt;sup>6</sup>Community Research Service Group, University of Shiga Prefecture, 2500 Yasakamachi, Hikoneshi, Shiga, 522-8533, Japan.

## **1** Introduction

The westerly wind transports mineral particles from the middle desert areas of the Asian continent, including the Gobi and Taklimakan Deserts, and mineral particles contaminated by anthropogenic pollutants at continental coasts are dispersed eastward over the Sea of Japan to the Japanese mainland (Duce et al., 1980; Iwasaka et al., 1983; Watanabe et al., 2006;

- Huang J. et al., 2015; Huang Z. et al., 2015; Huang et al., 2017a). In addition to abiotic particles, the microbial fractions associated with mineral-dust particles, which are commonly known as "bioaerosols", include viruses, bacteria, fungi, and pollen as well as plant and animal debris (Jones and Harrison, 2004; Jaenicke, 2005; Iwasaka et al., 2009; Pointing and Belnap, 2014). Asian dust events remarkably change the airborne microbial communities in the north-western part of China, which is close to the Gobi Desert (Tang et al., 2017). Moreover, the airborne bacterial compositions at high altitudes above
- Asian dust deposition and areas of anthropogenic pollution, such as Beijing (Li et al., 2010), Osaka (Yamaguchi et al., 2012), the Noto Peninsula (Maki et al., 2010, 2013), and the North American mountains (Smith et al., 2012), vary significantly. Airborne bacteria at high altitudes over East Asia are the focus of this study, because of their impacts on atmospheric chemical reactions and cloud formation (Pratt et al., 2009; Morris et al., 2011; Hara et al., 2016ab). In Japan, airborne microbial abundances increase in response to atmospheric depressions, which travel from the Asian continent (Murata and Content).
- Zhang, 2016; Sugimoto et al., 2012). Since desertification of dryland was thought to increase the transport of bioaerosols (Huang et al., 2017ab), the ecological impact on bioaerosols should be assessed. Some airborne microorganisms act as ice nuclei that are related to ice-cloud formation processes, indicating the possibility that wind-blown bioaerosol particles indirectly contribute to atmospheric radiation transfer and the geochemical cycle of atmospheric constituents (Möhler et al., 2007; Delort et al., 2010; Creamean et al., 2013; Joly et al., 2013; Pöschl and
- Shiraiwa, 2015). In particular, the ice-nucleating cell components of some bacterial species belonging to the phylum Proteobacteria and class Bacilli exhibit high nucleation activities, initiating ice formation at relatively warmer temperatures (from -5 °C to -2 °C) (Morris et al., 2004) than the inorganic ice-nuclei, such as potassium feldspar (approximately -8 °C) (Atkinson et al., 2013). Ice-nucleating bioaerosols are believed to activate ice-cloud formation more efficiently than inorganic substances (Hoose and Möhler, 2012; Murray et al., 2012), and contribute to rapid ice-cloud formation, even at
- low concentrations, in the clouds at temperatures between -8 °C and -3 °C (Hallett and Mossop, 1974). Airborne bacteria carried by westerly winds over East Asia were also found to initiate high levels of ice nucleation (Hara et al., 2016a, b). Culture-independent analyses for bacterial taxonomic composition demonstrated high bacterial diversity in cloudy waters, suggesting that some unculturable bacterial populations, including the keystone bacteria, primarily influence ice-cloud formations in North American mountains (Bowers et al., 2009; Pratt et al., 2009). However, in East Asia where
- desertification enhances the frequency of dust-events (Huang et al., 2017ab), the influences of airborne bacterial dynamics on atmospheric ice-nucleation and precipitation are unclear.

During the winter and early spring, strong north-westerly winds carry heavy falls of snow to Mt. Tateyama (3,015 m above sea level), which faces the East Sea. The snowfall sometimes includes natural and anthropogenic dust particles from the

Asian continent. Since ice-nucleating bacteria possibly contribute to ice-cloud formation and snow fall over Mt. Tateyama, the bacterial communities in the snow cover can be useful for investigating the impact of airborne bacteria on ice-cloud formation processes. The snow covers on Mt. Tateyama exhibit depths ranging from 6 m to 10 m in the spring (Osada et al., 2004; Watanabe et al., 2011). The air temperature, which rarely exceeds freezing from November to April, generally

- maintains the frozen condition of the snow cover until early April. Moreover, the snow cover located at high altitude avoids windblown contamination by regional soil materials. Previous researchers demonstrated that chemical compounds (Osada et al., 2004; Watanabe et al., 2011) and bioaerosols (Maki et al., 2011; Tanaka et al., 2011) from continental areas were found in the dirty layers of snow cover on Mt. Tateyama. Therefore, snow samples that include aerosols from continental areas can be obtained from the snow cover on Mt. Tateyama to analyse the airborne microbial communities relating to ice-nucleation
- at high altitudes.

This study investigated the ice-nucleation activities of intercontinentally transported aerosols in the snow cover, and identified the airborne bacterial changes relating to ice nucleation. Firstly, we dug a snow wall with a height of 734 cm in the snow cover on Mt. Tateyama in the early spring (April) and collected snow samples, including aerosols, from it. The snow samples were used for estimating the concentrations of chemical components and aerosol particles, using chemical analyses

and fluorescence microscopic observations, respectively. The ice-nucleation activities of aerosol particles in the snow samples were also evaluated by water-drop freezing assays. The bacterial community structures in snow samples were determined using MiSeq sequencing analysis using PCR-amplified bacterial 16S ribosomal RNA (rRNA) genes in order to investigate links between ice-nucleation activities and the vertical distribution of bacterial taxa in the snow cover.

#### 20 2. Sampling and Methods

#### 2.1 Snow sampling

The snow samples were collected from the snow cover at Murododaira  $(36.57^{\circ} \text{ N}, 137.60^{\circ} \text{ E}; 2,450 \text{ m})$  on Mt. Tateyama on the 20th April 2013 (Fig. 1). We dug a snow pit from the top of the snow-cover to the surface of the ground (743 cm vertical extent) and carefully smoothed the snow wall in the pit to leave the stratigraphy of the snow layers undisturbed. After the

- surface snow was removed from the snow wall using a sterilized snow sampler (polycarbonate plates: 3 cm × 20 cm × 0.1 cm), a 20 mL sample of snow was collected at a depth of 10 cm from the surface of the snow-wall using a new sterilized snow sampler. The snow samples were obtained from each 3 cm layer of the snow wall at lower heights from 164 cm to 200 cm (lower part) and upper heights from 560 cm to 743 cm (upper part). The upper parts frequently included the dirty layers with natural and/or anthropogenic dust particles, whereas the lower part was mainly composed of white layers (non-dirty
- layers). A total of 70 snow samples were obtained for use in fluorescence-microscopic observations, water-drop freezing assays (ice-nucleation assay) and DNA sequencing analyses (Fig. 2). Alternative snow samples were also collected from

each 10 cm layer from the top to the bottom of the snow wall for chemical analyses. The snow samples were preserved at -30 °C, prior to their use in each experiment.

#### 2.2 Chemical analyses, particle counts, and dust-event dating

The snow samples collected from each 10 cm layer were allowed to melt in the laboratory, and their chemical compositions 5 (anions and cations) were measured using ion chromatography (Dionex, ICS-1600 Thermo Fisher Scientific, Yokohama, Japan) (Watanabe et al., 2012). The values of nss-Ca<sup>2+</sup> and nss-SO<sub>4</sub><sup>2-</sup> were calculated from the concentration of Na<sup>+</sup> to Ca<sup>2+</sup> and SO<sub>4</sub><sup>2-</sup>, respectively. The accuracy of the measured values was approximately 5%. The concentrations of formaldehyde (HCHO) and acetaldehyde (CH<sub>3</sub>CHO) were determined using high-performance liquid chromatographic analysis (HPLC, LC-2000Plus, JASCO, Tokyo, Japan) with a fluorescent derivatizing reagent, such as 1, 3-cyclohexanedione (Iwama et al.,

- 2011; Watanabe et al., 2012). The detection limits of HCHO and CH<sub>3</sub>CHO were 0.05 and 0.01 µmol kg<sup>-1</sup>, respectively. The 500 µL solution of 70 snow samples collected from each 3-cm layer was fixed with a paraformaldehyde solution at a final concentration of 1 %. The samples were stained with DAPI (4',6-diamino-2-phenylindole) at a final concentration of 0.5 µg mL<sup>-1</sup> for 15 min and filtered through a 0.22 µm pore-size polycarbonate filter (Millipore, Tokyo, Japan) (Russell et al., 1974). After the filter was placed on a slide on top of a drop of low-fluorescence immersion oil, a drop of oil was added and
- then covered with a cover slide. Slides were examined using an epifluorescence microscope (Olympus, Tokyo, Japan) with UV excitation system. A filter transect was scanned, and the mineral particles (white particles), organic particles (yellow particles), bacterial cells (blue particles), and black carbon (black particles) on the filter transect were counted. In addition, the filter transect could be discriminate between yellow and white particles in two size categories of <5 μm and >5 μm. The attenuated backscatter coefficient and depolarization ratio measured by the light detection and ranging (LIDAR) system
- at Toyama (http://www-lidar.nies.go.jp/), which is located at a distance of 50 km from Mt. Tateyama, were used for evaluating the occurrences of dust events and anthropogenic pollutants, respectively. The LIDAR system was operated by AD-Net (Asian dust and aerosol lidar observation network) (Shimizu et al., 2016).

#### 2.3 Water-drop freezing assay

Twenty mL of melted snow samples (70 samples) of each 3-cm layer were passed through sterilized 0.22  $\mu$ m pore size membrane filters (Millipore, Billerica, MA, USA) to collect particulate matter. The particulate matter on the filters was resuspended in 1.0 mL of sterile nano-purewater. Re-suspended samples were diluted using the nano-purewater at the dilutions of some folds ranging from 0.1 to 20 and adjusted to the total particulate densities of approximately  $5.0 \times 10^5$  particles mL<sup>-1</sup> (from 1.0  $\mu$ g mL<sup>-1</sup> to 2.0  $\mu$ g mL<sup>-1</sup>). The total particulate densities were confirmed using fluorescence microscopic observations with DAPI staining. Fifty  $\mu$ L of the liquid was aliquoted into each of 24 wells in a sterile 96-well microplate.

- For the first assay, the 96-well microplates were placed onto an arminum plate and the measured temperature was decreased from 0 °C to -25 °C at a rate of 1.0 °C min<sup>-1</sup>. For each assay, wells of Arizona test dust (ATD: 2.0  $\mu$ g mL<sup>-1</sup>) and nano-
  - 4

purewater were prepared as positive and negative controls, respectively. Cumulative IN (Ice Nuclei) concentrations in 1 mL of melted snow at each temperature were calculated using the following equation (Vali, 1971).

$$IN = \frac{ln(N_{total}) - ln(N_{unfrozen})}{CV}$$

where N<sub>total</sub> is the total number of tubes (24 wells), N<sub>unfrozen</sub> is the number of wells still unfrozen (liquid) at each temperature,
C is the concentration ratio of melted snow ranging from 0.1 to 20, and V is the volume of melted snow (0.05 mL). In the present study, the measurable ice-nucleic concentrations in melted snow ranged from 174 IN L<sup>-1</sup> to 99,400 IN L<sup>-1</sup>. The snow samples including high concentrations of particulates showed similar levels of IN activities between unfiltrated solutions and particulate re-suspending solutions. The melted snow samples without filtration and with re-suspension showed similar ice-nuclei activities, demonstrating that the influence of soluble substrates on ice-nuclei activities can be neglected in this study.

## 10 2.4 High-throughput sequencing of bacterial 16S rRNA genes in the snow samples

The particles in 5 mL melted snow samples collected from 70 layers were pelleted by centrifuging at 20,000 g for 10 minutes and re-suspended into 500  $\mu$ L of nano-purewater. The re-suspending solutions were used for the extraction of genomic DNA (gDNA) using a phenol-chloroform method, which were combined with the microbial-cell degradation by SDS, proteinase K, and lysozyme, as described previously (Maki et al., 2008). Fragments of 16S rRNA gene (approximately 290 bp) were

- amplified from the extracted gDNA by PCR using universal bacterial primers 515F and 806R for the V4 region (Caporaso et al., 2011). The first PCR fragments were amplified again using the second PCR primers, which targeted the additional sequences of first PCR primers and included eight tag nucleotides designed for sample identification barcoding. Thermal cycling conditions were employed from the previous investigation (Maki et al., 2016). The PCR amplicons were used for high-throughput sequencing on a MiSeq Genome Sequencer (Illumina, CA, USA). The paired-end sequences with an area
- length of 250 bp were grouped based on the tag sequences for each sample. At the PCR-analysis steps, negative controls (nano-purewater) contained no fragments of 16S rRNA gene amplicons showing the absence of artificial contamination. The forward and reverse paired-end reads in the raw sequencing database were merged using USEARCH v.9.0.2132 (Edgar, 2013). After the irregular merged reads (lengths outside 200-500 bp range or exceeding 6 homopolymers) were removed by Mothur v1.36.1 (Schloss et al., 2009), sequences with low Q-scores (>1 expected error) and singleton reads were removed.
- These sequences were clustered *de novo* (with a minimum identity of 97 %) into operational taxonomic units (OTUs). The representative OTU sequences were identified using the RDP classifier (Wang et al., 2007) implemented in QIIME v9.1.1 (Caporaso et al., 2010). Greengenes release 13\_8 (McDonald et al., 2012) was used for determining taxonomic compositions. All sequences have been deposited in the DDBJ database (accession number of the submission is PRJEB24035).

#### 2.5 Quantitative real-time PCR (qRT-PCR)

A qRT-PCR analysis was employed, following the method of previous researchers (Kobayashi et al., 2015a), for investigating the relative abundance of bacteria through amplification of their 16S rRNA gene. Standard curves were

obtained using an ABI 7500 system (ABI, CA, USA), and calibrated by the several dilutions of purified bacterial amplicons. All the standard curves obtained in this way met the required standards of efficiency (R > 0.99, E > 90%). Reactions were performed in a 20-µL reaction mixture containing 10 µL of TaqMan Gene Expression Master Mix, 0.8 µL of Primer F20 (10 pmol µL<sup>-1</sup>), 0.8 µL of Primer R20 (10 pmol µL<sup>-1</sup>), 0.4 µL of TaqMan probe (10 pmol µL<sup>-1</sup>), and 2 µL of DNA template (10 ng mL<sup>-1</sup>), and 6.0 µL of nano-purewater. Amplification consisted of initial denaturation at 95 °C for 5 min, then 50 cycles at

95 °C for 15 s, followed by annealing and extension at 59 °C for 60 s. All reaction steps were performed using the ABI 7500

5

system.

#### 3. Results and Discussion

## 10 3.1 Vertical distributions of chemical compounds and particles in snow covers

The majority of snow samples recovered from Mt. Tateyama, using the method outlined in Section 2.1, were collected from the snow wall, of which most layers were composed of compacted snow or solid-type snow. The chemical compounds in the snow samples retained the variations corresponding to the snow layers, meaning that the snow layers would generally retain their original chemical and isotopic composition from the time of the snowfall (Fig. 3). The snow-cover layers were mainly composed of compacted snow (rounded grains), indicating that the snow samples maintained the records on atmospheric aerosols that were present during deposition and that they avoided melting. Some dirty layers that appeared brown-yellow or dark brown could be observed in four parts of the snow wall, at heights ranging from 599 cm to 620 cm, from 626 cm to 644 cm, from 650 cm to 662 cm, and from 716 cm to 734 cm. The solutions of snow samples from the dirty layers contained

higher concentrations of nss-Ca<sup>2+</sup>, which is a tracer of dust mineral particles from Asian deserts, than non-dirty layers in the

- lower parts of the snow wall. The nss-Ca<sup>2+</sup> peaks at each of the four dirty layers had maximum concentrations of 25.4 μeq L<sup>-1</sup>, 47.9 μeq L<sup>-1</sup>, 25.4 μeq L<sup>-1</sup>, and 31.4 μeq L<sup>-1</sup>, respectively (Fig. 3). Highly alkaline Ca is a tracer of mineral dusts from deserts and loess deposits in China (Suzuki and Tsunogai 1993). The deporalization ratio of LIDAR measurements also indicated the four series of Asian dust events that occurred from 12 to 21 February, from 26 February to 13 March, from 18 to 26 March and from 4 to 19 April in 2013 over the north-western seashore of the main island of Japan (Fig. 4, Fig. S1),
- corresponding to the four dirty layers found in the snow wall. The aerosols, which are transported from the Asian continent to Japan by the westerly wind, accumulate in the snow cover on Mt. Tateyama from fall to spring (Osada et al., 2004). The air back-trajectory analyses indicated that, during these periods, air masses came from the desert areas of the Asian continent (Fig. S2). In addition, the concentrations of NO<sub>3</sub><sup>-</sup>, SO<sub>4</sub><sup>2-</sup> and acetaldehyde, which are originate from the anthropogenic pollutions in the continental coastal area, also tended to increase in the snow samples collected from the dirty layers (Fig. 3).
- The chemical analyses in previous studies also detected anthropogenic pollutants in the snow cover of Mt. Tateyama. The snow samples of dirty layers significantly included high concentrations of acetaldehyde, which would have been synthesised from organic pollutants (Iwama et al., 2011; Watanabe et al., 2012). The natural dust events from desert areas accumulate

anthropogenic pollutants across the industrial areas during long-range transport processes (Huang J. et al., 2015). The dirty layers in snow wall would be formed during the four series of dust events during spring (March and April) and include intercontinentally transported aerosols that originated from the desert areas and industrial coasts of the Asian continent.

In contrast, the concentrations of formaldehyde photo-synthesised from plant products fluctuated in the range of 0.05 µmol

- 5 kg<sup>-1</sup> to 0.60 μmol kg<sup>-1</sup>, regardless of whether the samples were collected from non-dirty or dirty layers. Relatively constant high concentrations of around 0.45 μmol kg<sup>-1</sup> were observed in samples from the lower parts (from 167 cm to 200 cm) of the snow wall (Fig. 3). The concentrations of Na<sup>+</sup>, which primarily came from sea salt, increased in the snow samples in the two layers below the dirty layers (from 695 cm to 710 cm and from 584 cm to 589 cm) and in the low parts (from 167 cm to 200 cm) and ranged from from 14.9 μeg L<sup>-1</sup> to 28.9 μeg L<sup>-1</sup>. LIDAR measurements showed low ratios of deporalization and an
- attenuated backscatter coefficient before the middle of February 2013, indicating a lack of dust events during winter, when the snow cover from 167 cm to 200 cm formed (Fig. S1). The air back-trajectories before the middle of February 2013 frequently came from Siberia (Russia), North Asia, and the Sea of Japan, and remained over mountains and marine areas for longer periods, than those of April and March (Fig. S2). Airborne formaldehyde is not only synthesised by anthropogenic pollutants (Watanabe et al., 2012) but also photo-synthesised from plant products, such as isoprene (Claeys et al., 2004;
- Guenther et al., 2006). Formaldehyde detected in the lower parts of the snow wall was possibly transported from the mountains. The level of Na<sup>+</sup> of marine origin also increased in the lower parts, suggesting contamination by sea salts over the Sea of Japan. Consequently, we inferred that the aerosols recorded in the lower parts of the snow wall mainly came from Siberia (Russia), North Asia, and the Sea of Japan, whereas those in the upper parts, which frequently included Asian dust particles, originate from the desert regions and industrial coasts of the Asian continent.
- Mineral particles, yellow particles and bacterial particles were observed in the solutions of snow samples under fluorescent microscopic observation, using the DAPI staining technique. The total densities of DAPI-fluorescent particles increased in the snow samples of the four dirty layers and exhibited the peaks ranging from 6.33 x 10<sup>6</sup> to 4.12 x 10<sup>7</sup> particles mL<sup>-1</sup> (Fig. 5). In particular, yellow particles and bacterial particles in the snow samples from the two dirty layers had significantly higher densities, of the order of 10<sup>7</sup> particles mL<sup>-1</sup>. Black particles were frequently detected, regardless of whether a sample
- had been collected from a dirty layer, and the densities fluctuated by approximately 1.00 x 10<sup>5</sup> particles mL<sup>-1</sup>. Bacterial densities determined using qRT-PCR gradually increased from the lower parts to the upper parts of the snow wall and exhibited some peaks of the order of 10<sup>5</sup> copies mL<sup>-1</sup> in the dirty layers. The non-dirty snow samples from 698 cm to 695 cm also indicated high concentrations of DAPI-fluorescent particles and qRT-PCR detected bacteria (Fig. 5). Moreover, the concentrations of Ca<sup>2+</sup> and nitrate also increased in this layer (Fig. 3). This layer could be judged as not dirty, but some particles transported from the continent would be captured in this layer for short periods of time. This included Asian dust
- events, which have been reported to carry airborne microorganisms associated with natural mineral particles (Hara and Zhang, 2012) and anthropogenic particles on hazy days (Wei et al., 2016), leading to an increase in the microbial biomass in the downwind areas (Maki et al., 2014). The yellow fluorescence particles are organic materials that are interpreted to originate from dead microbial cells (Mostajir et al., 1995, Liu et al., 2014). Hara and Zhang (2012) reported that dust events
  - 7

in Kyushu, Japan, increased the ratio of damaged microbial cells in airborne microbial communities. Microbial cells transported by dust events would be exposed to environmental stressors throughout the atmosphere, increasing the number of damaged and dead cells in the dirty layers.

## 3.2 Ice-nucleation activities of snow samples in snow covers

- The ice-nucleation assay using some of the snow samples collected from the upper parts of the snow wall showed a higher freezing temperature of water drops than those from the lower parts of the snow wall (Fig. 6a). In particular, the snow samples of dirty layers (from 599 cm to 620 cm, from 626 cm to 644 cm, from 650 cm to 662 cm, and from 716 cm to 734 cm) indicated high freezing temperatures at more than -12 °C. The freezing temperatures showing the half concentrations of ice-nuclei particles (IN-T50C) are higher in the snow samples that include high amounts of fluorescent particles (Fig. 6b).
- The ice-nuclei particles detected in this study varied from 210 particles L<sup>-1</sup> to 442,000 particles L<sup>-1</sup> and were included in the range that the previous investigations estimated using rain and snow samples (Fig. 7) (Petters and Wright, 2015). At the freezing temperatures of -12 °C, the ice-nuclei particles in most of dirty-layer samples increased to more than 10,000 particles L<sup>-1</sup>, while most of the non-dirty layer samples and low-layer samples showed undetectable concentrations. The ice-nuclei activities in the snow samples are expected to change in accordance with the characteristics of organic and inorganic
- matter associated with aerosols. Dust mineral particles without organic matter, such as ATD, showed lower temperatures (below -15 °C) for the initial freezing of water drops than snow samples of the dirty layers. Additionally, the freezing temperatures of snow samples increased significantly in relation to the higher densities of fluorescent particles (P < 0.001) (Table 1). This means that the snow samples including dust and microbial particles have high activities of ice nucleation. Organic aerosols in the natural atmosphere are frequently reported to have higher activities of ice nucleation than inorganic
- particles (Hoose and Möhler, 2012; Murray et al., 2012). During the spring season, airborne microorganisms associated with Asian dust events possibly increase the ice-nucleation activities in the atmosphere and thus the amount of snow falling over Mt. Tateyama. For the investigation of the detailed characteristics of bacterial ice nucleation, a total of 11 isolates were obtained from the snow samples and their ice-nucleation activities were estimated using the water-drop freezing assay (Fig. S3). These bacterial isolates showed lower activities of ices nucleation than the snow samples of the dirty layers. In general,
- culturable microorganisms occupied 1%-10% of the environmental microbial communities (Olsen and Bakken 1987). Accordingly, the unculturable bacteria are expected to activate the majority of ice nucleation in the snow samples of dirty layers.

#### 3.3 Analyses of prokaryote community structures

For the analysis of bacterial compositions in the snow samples, we obtained a total of 17,676,926 reads. Following quality 30 filtering 6,367,300 merged paired-end sequences with a median length of 292 bp remained. The sequences of 16S rRNA gene were divided into 1,451 phylotypes (sequences with >97% similarity). Phylogenetic assignment of sequences resulted in an overall diversity comprising 33 phyla, (and candidate divisions), 73 classes (and class-level candidate taxa), and 179

families (and family-level candidate taxa). Most of the phylotypes recovered from the air samples were related to the phyla Cyanobacteria, Actinobacteria, Firmicutes (Bacilli, Clostridia), Bacteroidetes and Proteobacteria (Alpha-, Beta- and Gamma-proteobacteria), which are typically well represented in the 16S rRNA genes sequencing database generated from terrestrial, marine, freshwater, and phyllospheric environments (Fig. 8). Asian dust events dynamically change bacterial community

- 5 structures at high altitudes ranging from 200 m to 3,000 m above the ground (Jeon et al., 2011; Maki et al, 2013). For the PCR-analysis steps, negative controls (no template and a template from unused filters) did not contain the amplicons of 16S rRNA genes demonstrating the absence of artificial contamination during the experimental processes. The bacterial species numbers estimated by Chao 1 increased as the snow samples were collected from the upper parts of the
- snow wall (Fig. 9). The Chao 1 values were higher in the snow samples of dirty layers that those of non-dirty layers (P <</li>
  0.05). The rarefaction curve of all samples showed that the bacterial OUT numbers are mostly saturated at the numbers of determined sequences (data not shown) suggesting the sequencing database could follow the entire structure of bacterial communities. On a non-metric multidimensional scaling plot with weighted UniFrac distances, the snow samples can be categorised into three clusters, corresponding to the heights of the snow wall; samples from the lower parts (from 167 cm to 200 cm); samples from 563 cm to 668 cm; and samples from 671 cm to 734 cm (Fig. 10), indicating the variations of
- 15 bacteria with respect to the height of the snow cover above the base. The snow samples from the three dirty layers (February, March) overlapped with each other in the cluster of samples from 563 cm to 668 cm, whereas those of the other dirty layer (April) formed a different cluster of samples from 671 cm to 734 cm. The bacterial community structures in the clusters started at the low parts of the snow wall, moved to other areas on coordinate, and then returned to the original areas again, resulting in a "rotation" on the coordinate (indicated by arrows in Fig. 10). During the spring season, the variations in the
- 20 taxonomic composition (terrestrial, marine or plant-associated bacteria) of the airborne bacterial populations that accumulated on the snow cover would correspond to the heterogeneous mixtures of dust events responsible for transporting the bacteria. Presumably, the airborne bacterial compositions in snow covers at Mt. Tateyama are influenced by the intercontinentally transported aerosols.

#### 3.4 Distribution of prokaryote community structures in snow covers

Firmicutes (Bacilli) sequences, mainly belonging to the family Bacillaceae and Staphylococcuseae (>99.7 % similarity), dominated the snow samples of dirty layers, at relative abundances ranging from 25.0 % to 89.8 % (Fig. 8). The relative abundance of Firmicutes (Bacilli) sequences showed positive relations to the white-particle densities in snow sample, as well as ice-nucleation activities (P < 0.01) (Table 2). The Alpha and Betaproteobacteria sequences maintained high relative abundances ranging from 32.0 % to 83.9 % in the lower parts of the snow wall (lower than 590 cm: the winter season accumulation) and negatively related to the white-particle densities and the ice-nucleation activities. The relative abundances of Bacillaceae sequences showed a significant positive relationship with the ice-nucleation activities of the snow samples, whereas Proteobacteria showed a negative correlation to the ice-nucleation activities (Table 2). Some isolates of Bacilli obtained from cloud waters were confirmed to activate ice nucleation and Bacilli members have been focused on ice nuclear</p>

agents (Matulova et al., 2014; Mortazavi et al., 2015). Several members of Proteobacteria isolated from plant bodies showed high activities of ice-nucleation activities in laboratory experiments (Morris et al., 2004). The snow samples of dirty layers exhibited higher diversities of bacterial compositions than those of other layers, meaning that the dirty layers included a high numbers of minor bacterial species. We cannot neglect that the minor members also have high levels of IN activities, because entire heterogeneous bacterial communities would work as ice nuclei in this cold freezing assay.

In Proteobacteira sequences, the relative abundances of Oxalobacteraceae and Sphingomonadaceae sequences predominantly increased in the snow samples from the lower parts and showed significant positive relations to the Na<sup>+</sup> concentrations (Table 2) (P < 0.01). The Alpha-proteobacteria (in particular Sphingomonadaceae) (Cavicchioli et al., 2003), which predominately occupy marine bacterial communities, were also detected in the lower parts of snow wall (Fig. 8). The dust

particles transported across the Sea of Japan would be mixed with marine bacterial populations as well as sea salts (Zhang et al., 2006), thereby contributing to the marine bacterial transportation to Mt. Tateyama. Marine bacteria are frequently prevalent in the troposphere when strong onshore winds prevail (DeLeon-Rodriguez et al., 2013; Polymenakou at al., 2008; Maki et al., 2017).

The snow samples collected from the lower parts of the snow wall (winter-season accumulation) predominantly contained

- Proteobacteria sequences (Fig. 8), which were related to the dominant bacterial populations in the phyllosphere (Redford et al., 2010; Fierer and Lennon, 2011) or freshwater environments (Nold and Zwart, 1998). Proteobacteria members were frequently detected from the air samples collected over mountains (Bowers et al., 2012) or over the Noto Peninsula during periods when a north-westerly wind was blowing (Maki et al., 2010). These results suggest that some populations of Proteobacteria on Mt. Tateyama originated from the forest, rivers or lake areas of Siberia (Russia) or North Asia.
- The relative abundances of Actinobacteria sequences appeared randomly at low rates in the snow samples and slightly increased in the lower parts of the snow wall. In particular, Propionibacteriaceae sequences in the phylum Actinobacteria correlated positively to black-particle densities (P < 0.05) (Fig. 8). The Actinobacteria group (in particular Propionibacteriaceae) were primarily detected from anthropogenic particles collected in Beijing, China (Cao et al., 2014). Natural dust particles from Asian desert areas are mixed vertically with anthropogenic pollutants over the Asian continental
- coasts (Huang J. et al., 2015). However, non-spore forming bacteria, such as Propionibacteriaceae members, would be damaged by atmospheric stressors and few remaining populations could arrive in Japan continuously during the winter and spring seasons. Long-range transportation would select some bacterial populations among several terrestrial bacteria associated with dust particles that originated from the central desert area in Asia or an agriculture field in continental anthropogenic areas.
- The concentrations of formaldehyde changed from 0.05 µmol kg<sup>-1</sup> to 0.60 µmol kg<sup>-1</sup> regardless of whether the layers were derty, and maintained relatively high values in the low layers from 164 cm to 200 cm (Fig. 3). The Bacillaceae and Cytophagaceae sequences were significantly dominant in the snow samples collected from the dirty layers (Fig. 8) and their relative abundances positively correlated to the white-particle densities and the acetaldehyde concentrations (Table 2). Airborne acetaldehyde was reported to be photo-synthesised from the pollutant organic materials exposed to radiation from
  - 10

sunlight in the atmosphere (Iwama et al., 2011; Watanabe et al., 2012). Bacillaceae members were the dominant populations at high altitudes above the Taklimakan Desert (Maki et al., 2008) and Asian downwind areas during dust events (Korea: Jeon et al., 2011; Japan: Maki et al., 2014). The endospore forming bacteria, such as members of the family Bacillaceae, could survive for the duration of long-distance transit, because of their resistance to UV irradiation and desiccations in the

5 atmosphere (Kobayashi et al., 2015b). The Bacteroidetes sequences including Cytophagaceae were also often detected from the aerosols sampled at high altitudes during Asian dust events (Maki et al, 2013, 2015). Since Cytophagaceae members tend to aggregate with the organic particles in terrestrial and aquatic environments (Newton et al., 2011), the bacterial cells involved in aggregations would be protected against atmospheric stressors. Long-range transportation would select the stressor-resisting bacteria among several bacterial communities associated with dust events.

#### 10 4. Conclusion

15

30

The sequential changes of airborne bacteria from winter to spring have been investigated using aerosols that have been transported for long-distances and preserved in snow cover; their relations to ice-nucleation activities of snow samples were also evaluated. During the winter season, the north-westerly wind would transport members of the phylum Proteobacteria, which are expected to originate from phyllosphere, freshwater in mountainous areas, or marine environments. The intercontinental dust events during spring would carry the terrestrial bacteria of Bacilli (Bacillaceae) and Bacteroidets (Cytophagaceae) from the Asian continent to Mt. Tateyama, whereas other terrestrial bacteria of Actinobacteria mainly disappear across the Sea of Japan. Asian dust-associated bacteria, such as Bacilli, showed positive relations to ice-nucleation

activities in snow samples. Since the ice-nucleation activities of bacterial isolates from snow samples are lower than those of snow samples, unculturable bacteria in the snow samples are expected to be responsible for high levels of ice-nucleation

- 20 activities. Airborne microorganisms suspended over the Japanese islands might act as ice nuclei supporting the heavy snow falling over Mt. Tateyama during spring season. Furthermore, the characteristics of bioaerosols deposited onto the snow surface needed to be understood to highlight the impacts of microorganisms on surface albedo and the melting rate of snow. In the future, the combination of sequencing analysis, with physiological experiments targeting bacterial cultures, and metagenome analysis, targeting functional genes, would support to elucidate airborne bacterial influences on climate change
- 25 and human societies.

#### Acknowledgements

Members of Fasmac Co., Ltd. helped with the MiSeq sequencing analyses. LIDAR data operated by AD-Net were used for the identification of dust-event occurrences. This study was funded by the Joint Research Program of the Arid Land Research Center, Tottori University (No. 28C2015). This study was supported by the Grant-in-Aid for Scientific Research (A) (No. 17H01616), (B) (No. 26304003) and (C) (No. 26340049), the Bilateral Joint Research Projects from the Japanese

Society for the Promotion of Science (JSPS), and the Strategic Young Researcher Overseas Visits Program for Accelerating Brain Circulation (No. G2702).

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

10

5

15

| Tat               | ole 1. Relativ             | es of the DA   | PI-fluor   | resecent part  | ticle | s with ice-nuc      | leation activiti  | ies and some o  | of chemical con      |  |  |
|-------------------|----------------------------|----------------|------------|----------------|-------|---------------------|-------------------|-----------------|----------------------|--|--|
|                   |                            | Ice-nuclea     | tion activ | vities**       |       | Chemical components |                   |                 |                      |  |  |
| DAPI-fluoresecent | pa<br>Initial              | temp.End       | temp.      | IN-T50C        |       | Form-<br>aldehyde   | Acet-<br>aldehyde | Nå              | nss-Ca <sup>2+</sup> |  |  |
| Yellow ≥5 µm      | <b>0.34</b> †              | 0.4            | 8††        | <b>0.54</b> †† |       | -0.13               | <b>0.40</b> ††    | -0.23           | <b>0.31</b> †        |  |  |
| Yellow <5 µm      | <b>0.47</b> †*             | † <b>0.6</b> : | 5††        | <b>0.69</b> †† |       | -0.06               | <b>0.42</b> ††    | - <b>0.34</b> † | <b>0.44</b> ††       |  |  |
| White ≥5 µm       | <b>0.49</b> †*             | † 0.5          | 9††        | 0.69††         |       | -0.12               | 0.50††            | <b>-0.40</b> †† | <b>0.39</b> ††       |  |  |
| White <5 µm       | <b>0.38</b> † <sup>-</sup> | † 0.5          | 1††        | 0.57††         |       | -0.05               | 0.39††            | -0.34†          | <b>0.42</b> ††       |  |  |
| Blue              | <b>0.43</b> †              | † 0.5:         | 5††        | 0.62††         |       | -0.04               | 0.35††            | <b>-0.41</b> †† | <b>0.42</b> ††       |  |  |
| Black             | 0.13                       | 0.3            | 6††        | 0.32†          |       | 0.08                | 0.11              | 0.00            | 0.26                 |  |  |

\* The marks †† indicates P<0.001 and the mark † indicates P<0.005. Among the marked values, red cells indicate positive r relations.

\*\* Initial temp., End temp., and IN-T50C indicate the initial-freezing temperatures of water drops, the end-freezing temperatures of N particle concentrations, respectively.

| Bacterial categorial Yellow         Yellow <th></th> <th></th> <th></th> <th>Fluorescent p</th> <th>article conce.</th> <th>ntrations</th> <th></th> <th></th> <th></th> <th>Ч<sup>р</sup></th> <th>emical com</th> <th>onents</th> <th></th> <th></th> <th></th> <th></th>   |                           |                            |                 | Fluorescent p  | article conce. | ntrations   |                 |                   |                   | Ч <sup>р</sup> | emical com          | onents        |         |                     |           |           |
|-------------------------------------------------------------------------------------------------------------------------------------------------------------------------------------------------------------------------------------------------------------------------------------------------------------------------------------------------------------------------------------------------------------------------------------------------------------------------------------------------------------------------------------------------------------------------------------------------------------------------------------------------------------------------------------------------------------------------------------------------------------------------------------------------------------------------------------------------------------------------------------------------------------------------------------------------------------------------------------------------------------------------------------------------------------------------------------------------------------------------------------------------------------------------------------------------------------------------------------------------------------------------------------------------------------------------------------------------------------------------------------------------------------------------------------------------------------------------------------------------------------------------------------------------------------|---------------------------|----------------------------|-----------------|----------------|----------------|-------------|-----------------|-------------------|-------------------|----------------|---------------------|---------------|---------|---------------------|-----------|-----------|
| Cyanobacteria         0.03         -0.12         -0.13         -0.13         -0.13         -0.13         -0.13         -0.13         -0.13         -0.13         -0.13         -0.13         -0.13         -0.13         -0.13         -0.13         -0.13         -0.23         -0.23+1         -0.34+1         -0.26         0.01         -0.01         -0.01         -0.03         -0.34         -0.34+1         -0.35         -0.32+1         -0.34+1         -0.32         -0.34+1         -0.35         -0.34+1         -0.36         -0.01         -0.01         -0.01         -0.03         -0.34         -0.34         -0.34         -0.34         -0.34+1         -0.35         -0.34+1         -0.35         -0.34+1         -0.35         -0.34         -0.34         -0.34         -0.34         -0.34         -0.34         -0.34         -0.34         -0.34         -0.34         -0.34         -0.34         -0.34         -0.34         -0.34         -0.34         -0.34         -0.34         -0.34         -0.34         -0.34         -0.34         -0.34         -0.34         -0.34         -0.34         -0.34         -0.34         -0.34         -0.34         -0.34         -0.34         -0.34         -0.34         -0.34         -0.34         -0.34                                                                                                                                                                                                                                                                         | Bacterial categorie       | s Yellow<br>um             | Yellow<br>≪5 µm | White<br>um    | White<br>um    | Blue        | Black           | Form-<br>aldehvde | Acet-<br>aldehvde | N <sup>+</sup> | nss-Ca <sup>+</sup> | NH            | 00      | nss-SQ <sup>-</sup> | IN-T50C** | Heights** |
| Xittobacteria0.130.230.32440.3440.200.100.000.000.000.010.0340.3440.344Firmicates (Darrial)0.27*0.331*0.331*0.331*0.321*0.321*0.321*0.321*0.324*0.324*0.324*0.324*0.324*0.324*0.324*0.324*0.324*0.324*0.324*0.324*0.324*0.324*0.324*0.324*0.324*0.324*0.324*0.324*0.324*0.324*0.324*0.324*0.324*0.324*0.324*0.324*0.324*0.324*0.324*0.324*0.324*0.324*0.324*0.324*0.324*0.324*0.324*0.324*0.324*0.324*0.324*0.324*0.324*0.324*0.324*0.324*0.324*0.324*0.324*0.324*0.324*0.324*0.324*0.324*0.324*0.324*0.324*0.324*0.324*0.324*0.324*0.324*0.324*0.324*0.324*0.324*0.324*0.324*0.324*0.324*0.324*0.324*0.324*0.324*0.324*0.324*0.324*0.324*0.324*0.324*0.324*0.324*0.324*0.324*0.324*0.324*0.324*0.324*0.324*0.324*0.324*0.324*0.324*0.324*0.324*0.324*0.324*0.324*0.324*0.324*0.324*0.324*0.324*0.324*0.324*0.324*0.324*0.324*0.324*0.324*                                                                                                                                                                                                                                                                                                                                                                                                                                                                                                                                                                                                                                                                                                                  | Cyanobacteria             | 0.03                       | -0.12           | -0.10          | -0.17          | -0.15       | 0.19            | -0.01             | -0.10             | -0.08          | 0.06                | 0.12          | 0.08    | 0.14                | -0.10     | 0.06      |
| Financiane (Bacill)         0.27+         0.43+         0.43+         0.43+         0.43+         0.43+         0.43+         0.43+         0.43+         0.43+         0.43+         0.43+         0.43+         0.43+         0.43+         0.43+         0.43+         0.43+         0.43+         0.43+         0.43+         0.43+         0.43+         0.43+         0.43+         0.43+         0.43+         0.43+         0.43+         0.43+         0.43+         0.43+         0.43+         0.43+         0.43+         0.43+         0.43+         0.43+         0.43+         0.43+         0.43+         0.43+         0.43+         0.43+         0.43+         0.43+         0.43+         0.43+         0.43+         0.43+         0.43+         0.43+         0.43+         0.43+         0.43+         0.43+         0.43+         0.43+         0.43+         0.43+         0.43+         0.43+         0.43+         0.43+         0.43+         0.43+         0.43+         0.43+         0.43+         0.43+         0.43+         0.43+         0.43+         0.43+         0.43+         0.43+         0.43+         0.43+         0.43+         0.43+         0.43+         0.43+         0.43+         0.43+         0.43+         0.43+         0.44+ <td>Actinobacteria</td> <td>-0.13</td> <td>-0.23</td> <td>-0.32++</td> <td>-0.34 ††</td> <td>-0.26</td> <td>0.10</td> <td>0.16</td> <td>-0.36††</td> <td>0.16</td> <td>-0.07</td> <td>0.00</td> <td>-0.01</td> <td>0.0</td> <td>-0.34</td> <td>-0.24</td>             | Actinobacteria            | -0.13                      | -0.23           | -0.32++        | -0.34 ††       | -0.26       | 0.10            | 0.16              | -0.36††           | 0.16           | -0.07               | 0.00          | -0.01   | 0.0                 | -0.34     | -0.24     |
| Findentical (c) friding)         0.01         0.01         0.01         0.01         0.01         0.01         0.01         0.01         0.01         0.01         0.01         0.01         0.01         0.01         0.01         0.01         0.01         0.01         0.01         0.01         0.01         0.01         0.01         0.01         0.01         0.01         0.01         0.01         0.01         0.01         0.01         0.01         0.01         0.01         0.01         0.01         0.01         0.01         0.01         0.01         0.01         0.01         0.01         0.01         0.01         0.01         0.01         0.01         0.01         0.01         0.01         0.01         0.01         0.01         0.01         0.01         0.01         0.01         0.01         0.01         0.01         0.01         0.01         0.01         0.01         0.01         0.01         0.01         0.01         0.01         0.01         0.01         0.01         0.01         0.01         0.01         0.01         0.01         0.01         0.01         0.01         0.01         0.01         0.01         0.01         0.01         0.01         0.01         0.01         0.01         0                                                                                                                                                                                                                                                                                                        | Firmicutes (Bacilli)      | 0.27††                     | 0.33††          | 0.43††         | 0.32           | 0.36††      | -0.01           | -0.32++           | 0.43††            | -0.44††        | 0.20                | 0.33††        | 0.19    | 0.32†               | 0.35†     | 0.58††    |
| Backeroidetes         -0.05         0.00         -0.01         0.05         -0.05         -0.35         -0.35         -0.15         -0.21         -0.23         -0.23         -0.23         -0.23         -0.23         -0.23         -0.23         -0.23         -0.23         -0.23         -0.23         -0.23         -0.23         -0.23         -0.23         -0.23         -0.23         -0.23         -0.23         -0.23         -0.23         -0.23         -0.23         -0.23         -0.23         -0.23         -0.23         -0.23         -0.23         -0.23         -0.23         -0.23         -0.23         -0.23         -0.23         -0.23         -0.23         -0.23         -0.23         -0.23         -0.23         -0.23         -0.23         -0.23         -0.23         -0.23         -0.23         -0.23         -0.23         -0.23         -0.23         -0.23         -0.23         -0.23         -0.23         -0.23         -0.23         -0.23         -0.23         -0.23         -0.23         -0.23         -0.23         -0.23         -0.23         -0.23         -0.23         -0.23         -0.23         -0.23         -0.23         -0.23         -0.23         -0.23         -0.23         -0.23         -0.23         -0.23                                                                                                                                                                                                                                                                               | Firmicutes (Clostridia)   | -0.03                      | 0.16            | 0.12           | 0.14           | 0.13        | -0.28++         | 0.07              | -0.12             | -0.03          | -0.12               | 0.07          | 0.03    | -0.05               | 0.06      | 0.11      |
| Application         0.15         0.23H         0.33H         0.23H                                                                                                                                                                                                                                                                                | Bacteroidetes             | -0.05                      | 0.00            | -0.01          | 0.05           | -0.05       | -0.35††         | -0.16             | -0.09             | -0.25†         | -0.19               | -0.22†        | -0.20   | -0.22               | -0.18     | 0.24      |
| Betaproteobacteria         0.06         0.41+         0.41+         0.41+         0.31+         0.31+         0.41+         0.41+         0.31+         0.31+         0.42+         0.42+         0.42+         0.42+         0.42+         0.42+         0.42+         0.42+         0.42+         0.42+         0.42+         0.42+         0.42+         0.42+         0.42+         0.42+         0.42+         0.42+         0.42+         0.42+         0.42+         0.42+         0.42+         0.42+         0.42+         0.42+         0.42+         0.42+         0.42+         0.42+         0.42+         0.42+         0.42+         0.42+         0.42+         0.42+         0.42+         0.42+         0.42+         0.42+         0.42+         0.42+         0.42+         0.42+         0.42+         0.42+         0.42+         0.42+         0.42+         0.42+         0.42+         0.42+         0.42+         0.42+         0.42+         0.44+         0.44+         0.44+         0.44+         0.44+         0.44+         0.44+         0.44+         0.44+         0.44+         0.44+         0.44+         0.44+         0.44+         0.44+         0.44+         0.44+         0.44+         0.44+         0.44+         0.44+         0.44+ <td>Alphaproteobacteria</td> <td>-0.15</td> <td>-0.27++</td> <td>-0.33++</td> <td>-0.28††</td> <td>-0.25</td> <td>0.20</td> <td>0.18</td> <td>-0.27++</td> <td>0.42+†</td> <td>-0.13</td> <td>-0.24†</td> <td>-0.18</td> <td>-0.23</td> <td>-0.15</td> <td>-0.57††</td> | Alphaproteobacteria       | -0.15                      | -0.27++         | -0.33++        | -0.28††        | -0.25       | 0.20            | 0.18              | -0.27++           | 0.42+†         | -0.13               | -0.24†        | -0.18   | -0.23               | -0.15     | -0.57††   |
| Gammaprotecharteria         0.06         0.05         0.15         0.05         0.05         0.05         0.05         0.05         0.05         0.05         0.05         0.01         0.02         0.02         0.02         0.02         0.02         0.02         0.02         0.02         0.02         0.02         0.02         0.02         0.02         0.02         0.02         0.02         0.02         0.02         0.02         0.02         0.02         0.02         0.02         0.02         0.02         0.02         0.02         0.02         0.02         0.02         0.02         0.02         0.02         0.02         0.02         0.02         0.02         0.02         0.02         0.02         0.02         0.02         0.02         0.02         0.02         0.02         0.02         0.02         0.02         0.02         0.02         0.02         0.02         0.02         0.02         0.02         0.02         0.02         0.02         0.02         0.02         0.02         0.02         0.02         0.02         0.02         0.02         0.02         0.02         0.02         0.02         0.02         0.02         0.02         0.02         0.02         0.02         0.02         0.02 </td <td>Betaproteobacteria</td> <td>-0.26†</td> <td>-0.42++</td> <td>-0.44††</td> <td>-0.42††</td> <td>-0.41</td> <td>0.06</td> <td>0.35††</td> <td>-0.41 + +</td> <td>0.54††</td> <td>-0.31</td> <td>-0.31††</td> <td>-0.18</td> <td>-0.30†</td> <td>-0.25</td> <td>-0.75††</td>                       | Betaproteobacteria        | -0.26†                     | -0.42++         | -0.44††        | -0.42††        | -0.41       | 0.06            | 0.35††            | -0.41 + +         | 0.54††         | -0.31               | -0.31††       | -0.18   | -0.30†              | -0.25     | -0.75††   |
| Bacterial others         0.03         0.03         0.04         0.19         0.03         0.03         0.03         0.03         0.03         0.03         0.03         0.03         0.03         0.03         0.03         0.03         0.03         0.03         0.03         0.03         0.03         0.03         0.03         0.03         0.03         0.03         0.03         0.03         0.03         0.03         0.03         0.03         0.03         0.03         0.03         0.03         0.03         0.03         0.03         0.03         0.03         0.03         0.03         0.03         0.03         0.03         0.03         0.03         0.03         0.03         0.03         0.03         0.03         0.03         0.03         0.03         0.03         0.03         0.03         0.03         0.03         0.03         0.03         0.03         0.03         0.03         0.03         0.03         0.03         0.03         0.03         0.03         0.03         0.03         0.03         0.03         0.03         0.03         0.03         0.03         0.03         0.03         0.03         0.03         0.03         0.03         0.03         0.03         0.03         0.03         0.03                                                                                                                                                                                                                                                                                                             | Gammaproteobacteria       | 0.06                       | 0.05            | 0.15           | 0.05           | 0.07        | 0.07            | 0.03              | 0.05              | -0.04          | -0.01               | 0.19          | 0.16    | 0.13                | 0.02      | 0.22      |
| Micrococacae         -0.23t         0.34ti         0.35ti         0.35ti         0.35ti         0.35ti         0.35ti         0.20ti         0.00ti         0.20ti         0.00ti         0.20ti         0.00ti         0.20ti         0.00ti         0.21ti         0.01ti                                                                                                                                                                                                                                                   | <b>Bacterial others</b>   | 0.03                       | 0.23†           | 0.09           | 0.31           | 0.19        | 0.06            | -0.04             | 0.29††            | -0.10          | 0.47††              | -0.03         | 0.08    | 0.02                | 0.07      | 0.16      |
| Propionibacteriaces         0.05         0.05         0.01         0.03         0.01         0.24         0.21         0.24         0.06         0.24         0.06         0.06         0.06           Bacilaceac         0.16         0.17         0.314         0.15         0.03         0.016         0.03         0.04         0.16         0.17         0.031         0.02         0.03           Accooccaceac         0.09         0.14         0.05         0.13         0.214         0.216         0.012         0.012         0.012         0.029         0.035           Suphylococcaceac         0.16         0.13         0.214         0.216         0.217         0.217         0.216         0.012         0.012         0.012         0.012         0.02         0.02         0.02         0.02         0.02         0.02         0.02         0.02         0.02         0.02         0.02         0.02         0.02         0.02         0.02         0.02         0.02         0.02         0.02         0.02         0.02         0.02         0.02         0.02         0.02         0.02         0.02         0.02         0.02         0.02         0.02         0.02         0.02         0.02         0.02                                                                                                                                                                                                                                                                                                                        | Micrococcaceae            | -0.23†                     | -0.34††         | -0.37++        | -0.35††        | -0.29       | -0.05           | 0.21†             | -0.43††           | 0.38††         | -0.30†              | -0.29††       | -0.29   | -0.27               | -0.24     | -0.64††   |
| Bacillacae         0.16         0.17         0.31         0.15         0.31         0.15         0.15         0.15         0.15         0.15         0.15         0.15         0.15         0.15         0.15         0.15         0.15         0.16         0.11         0.12         0.02         0.03           Staphylococacae         0.09         0.14         0.06         0.13         0.214         0.215         0.02         0.12         0.02         0.02         0.23           Staphylococacae         0.15         0.01         0.16         0.17         0.02         0.17         0.02         0.13         0.29           Staphylococacae         0.13         0.214         0.214         0.23         0.20         0.04         0.17         0.02         0.13         0.29         0.29         0.29         0.29         0.29         0.29         0.29         0.29         0.29         0.29         0.29         0.29         0.29         0.29         0.29         0.29         0.29         0.29         0.29         0.29         0.29         0.29         0.29         0.29         0.29         0.29         0.29         0.29         0.29         0.29         0.20         0.21         0.21                                                                                                                                                                                                                                                                                                                            | Propionibacteriaceae      | 0.05                       | 0.05            | -0.01          | 0.03           | 0.10        | 0.24†           | -0.03             | -0.05             | -0.37++        | 0.21                | 0.24          | 0.22    | 0.44††              | -0.06     | 0.29      |
| Arrencoccateae         0.09         0.14         0.06         0.13         0.214         0.204         0.14         0.01         0.16         0.12         0.02         0.02         0.02           Staphylococcateae         0.15         0.20         0.18         0.214         0.224         0.17         0.026         0.17         0.02         0.17         0.13         0.23           Staphylococcateae         0.16         0.16         0.10         0.214         0.214         0.214         0.17         0.02         0.17         0.13         0.23           Cytophagaceae         0.04         0.15         0.01         0.05         0.17         0.02         0.17         0.13         0.24         0.13         0.23         0.03         0.24         0.13         0.13         0.23         0.24         0.24         0.13         0.21         0.13         0.23         0.24         0.24         0.21         0.23         0.24         0.24         0.24         0.23         0.24         0.24         0.23         0.24         0.24         0.21         0.23         0.24         0.24         0.21         0.23         0.24         0.24         0.24         0.23         0.24         0.21 <th< td=""><td>Bacillaceae</td><td>0.16</td><td>0.17</td><td>0.31</td><td>0.15</td><td>0.20</td><td>-0.16</td><td>-0.32++</td><td>0.49††</td><td>-0.31++</td><td>0.18</td><td>0.19</td><td>0.18</td><td>0.18</td><td>0.29</td><td>0.35†</td></th<>                                                                            | Bacillaceae               | 0.16                       | 0.17            | 0.31           | 0.15           | 0.20        | -0.16           | -0.32++           | 0.49††            | -0.31++        | 0.18                | 0.19          | 0.18    | 0.18                | 0.29      | 0.35†     |
| Staphylococcaceae         0.15         0.20         0.18         0.21         0.20         0.04         0.17         0.02         0.17         0.13         0.23           Cytophagaceae         0.04         0.15         0.00         0.16         0.10         0.27         0.02         0.18         -0.16         0.13         0.13         0.20         0.13         0.13         0.13         0.13         0.13         0.13         0.13         0.13         0.13         0.13         0.13         0.13         0.13         0.13         0.13         0.13         0.13         0.13         0.13         0.13         0.13         0.13         0.13         0.13         0.13         0.13         0.13         0.13         0.13         0.13         0.13         0.13         0.13         0.13         0.13         0.13         0.13         0.13         0.13         0.13         0.13         0.13         0.13         0.13         0.13         0.13         0.13         0.13         0.13         0.13         0.13         0.13         0.13         0.13         0.13         0.13         0.13         0.13         0.13         0.13         0.13         0.13         0.13         0.13         0.13                                                                                                                                                                                                                                                                                                                          | Aerococcaceae             | 0.09                       | 0.14            | 0.06           | 0.13           | $0.21 \div$ | -0.20           | 0.00              | -0.08             | -0.14          | 0.01                | 0.16          | 0.04    | 0.12                | 0.02      | 0.22      |
| Cytophagaceae         0.04         0.15         0.00         0.16         0.10         0.221         0.201         0.266         0.18         0.18         0.18         0.18         0.18         0.16         0.10           Chitinophagaceae         0.18         0.11         0.16         0.17         0.00         0.01         0.23         0.06         0.11         0.21         0.21         0.24         0.17         0.21         0.20         0.24         0.01         0.00         0.00         0.00         0.00         0.01         0.01         0.01         0.02         0.03         0.21         0.24         0.21         0.21         0.20         0.01         0.00         0.00         0.00         0.00         0.01         0.01         0.01         0.01         0.01         0.01         0.01         0.01         0.01         0.01         0.01         0.01         0.01         0.02         0.03         0.01         0.01         0.01         0.01         0.02         0.03         0.01         0.01         0.01         0.01         0.01         0.01         0.02         0.03         0.01         0.01         0.01         0.01         0.01         0.01         0.01         0.01 <td< td=""><td>Staphylococcaceae</td><td>0.15</td><td>0.20</td><td>0.18</td><td>0.21</td><td>0.22</td><td>0.15</td><td>-0.08</td><td>0.03</td><td>-0.20</td><td>0.04</td><td>0.17</td><td>0.02</td><td>0.17</td><td>0.13</td><td>0.29</td></td<>                                                                        | Staphylococcaceae         | 0.15                       | 0.20            | 0.18           | 0.21           | 0.22        | 0.15            | -0.08             | 0.03              | -0.20          | 0.04                | 0.17          | 0.02    | 0.17                | 0.13      | 0.29      |
| Chilinophagaceae $-0.18$ $-0.11$ $-0.16$ $-0.17$ $-0.23$ $-0.06$ $-0.11$ $-0.21$ $-0.21$ $-0.21$ $-0.21$ $-0.21$ $-0.21$ $-0.21$ $-0.21$ $-0.21$ $-0.21$ $-0.21$ $-0.21$ $-0.21$ $-0.21$ $-0.21$ $-0.21$ $-0.21$ $-0.21$ $-0.21$ $-0.21$ $-0.21$ $-0.21$ $-0.21$ $-0.21$ $-0.21$ $-0.21$ $-0.21$ $-0.21$ $-0.21$ $-0.21$ $-0.21$ $-0.21$ $-0.21$ $-0.21$ $-0.21$ $-0.21$ $-0.21$ $-0.21$ $-0.21$ $-0.21$ $-0.21$ $-0.21$ $-0.21$ $-0.21$ $-0.21$ $-0.21$ $-0.21$ $-0.21$ $-0.21$ $-0.21$ $-0.21$ $-0.21$ $-0.21$ $-0.21$ $-0.21$ $-0.21$ $-0.21$ $-0.21$ $-0.21$ $-0.21$ $-0.21$ $-0.21$ $-0.21$ $-0.21$ $-0.21$ $-0.21$ $-0.21$ $-0.21$ $-0.21$ $-0.21$ $-0.21$ $-0.21$ $-0.21$                                                                                                                                                                                                                                                                                                                                                                                                                                                                                                                                                                                                                                                                                                                                                                                                                                                                            | Cytophagaceae             | 0.04                       | 0.15            | 0.00           | 0.16           | 0.10        | -0.27           | -0.21             | -0.17             | -0.26          | -0.18               | -0.20         | -0.18   | -0.18               | -0.16     | 0.13      |
| Sphingomonadaceae $-0.37$ $-0.38$ $-0.34$ $0.03$ $0.17$ $0.33$ $-0.27$ $-0.26$ $-0.30$ $-0.20$ $-0.30$ Comamonadaceae $0.06$ $0.01$ $0.02$ $0.07$ $0.00$ $0.05$ $-0.31$ $-0.24$ $-0.34$ $-0.30$ $-0.20$ $-0.30$ Comamonadaceae $0.06$ $0.01$ $0.02$ $0.07$ $0.00$ $0.07$ $0.00$ $0.01$ $0.16$ $-0.31$ $-0.34$ $-0.34$ $-0.34$ $-0.32$ $-0.02$ $-0.03$ $0.16$ $0.16$ $0.16$ $0.16$ $0.01$ $0.01$ $0.12$ $-0.02$ $-0.03$ $0.00$ $0.16$ $0.06$ $0.16$ $0.06$ $0.01$ $0.02$ $0.03$ $0.01$ $0.02$ $0.03$ $0.01$ $0.02$ $0.03$ $0.06$ $0.03$ $0.06$ $0.03$ $0.06$ $0.03$ $0.06$ $0.03$ $0.06$ $0.03$ $0.06$ $0.03$ $0.06$ $0.03$ $0.06$ $0.03$                                                                                                                                                                                                                                                                                                                                                                                                                                                                                                                                                                                                                                                                                                                                                                                                                                                                                                                    | Chitinophagaceae          | -0.18                      | -0.18           | -0.11          | -0.16          | -0.17       | -0.07           | -0.23†            | -0.06             | -0.11          | -0.21               | -0.24†        | -0.17   | -0.21               | -0.27     | 0.06      |
| Commondaceae         0.06         0.01         0.02         0.07         0.07         0.06         0.07         0.07         0.01         0.12         -0.02         -0.09         0.16           Oxalobacteraceae         -0.34+         -0.51+         -0.41+         -0.42+         -0.02         -0.03         -0.03         0.01         0.12         -0.02         -0.09         0.16           Xanthomondaceae         -0.10         0.01         0.07         0.02         -0.02         -0.34+         -0.34+         -0.34         -0.34         -0.33         -0.37         -0.37         -0.37         -0.37         -0.37         -0.37         -0.37         -0.37         -0.37         -0.37         -0.37         -0.37         -0.37         -0.37         -0.37         -0.37         -0.37         -0.37         -0.37         -0.37         -0.37         -0.37         -0.37         -0.37         -0.37         -0.37         -0.37         -0.37         -0.37         -0.37         -0.37         -0.37         -0.37         -0.37         -0.37         -0.37         -0.37         -0.37         -0.37         -0.37         -0.37         -0.37         -0.37         -0.37         -0.37         -0.37         -0.37         -0.37                                                                                                                                                                                                                                                                                            | Sphingomonadaceae         | -0.26                      | -0.37++         | -0.38++        | -0.36††        | -0.34       | 0.05            | 0.17              | -0.33++           | 0.39††         | -0.22               | -0.27         | -0.26   | -0.30               | -0.20     | -0.59††   |
| Oxalobacteracee $-0.34\dagger$ $-0.45\dagger$ $-0.51\dagger$ $-0.11\dagger$ $-0.42\dagger$ $-0.22$ $-0.34\dagger$ $-0.26$ $-0.33\dagger$ $-0.24$ $-0.87\dagger$ Xanthomonadaceae $-0.10$ $0.01$ $0.07$ $0.07$ $0.09$ $-0.02$ $0.03$ $-0.01$ $-0.05$ $0.08$ $0.00$ $0.04$ $0.03$ $0.06$ Peudomonadaceae $0.18$ $0.00$ $0.02$ $-0.08$ $-0.01$ $-0.02$ $-0.07$ $0.06$ $0.04$ $0.03$ $0.06$ **The marks $\dagger$ indicates $0.00$ $0.02$ $-0.08$ $-0.10$ $0.17$ $0.01$ $-0.05$ $0.03$ $0.04$ $0.03$ $0.04$ **The marks $\dagger$ indicates $0.00$ $0.02$ $-0.08$ $-0.10$ $0.17$ $0.01$ $-0.05$ $0.03$ $0.04$ $0.03$ $0.04$ **The temperature indicating the 50% of concentrations of features $-1.10$ $0.17$ $0.01$ $-0.05$ $0.01$ $0.02$ $-0.08$ $-0.08$ $0.00$ **The heights (cm) of snow wall, from which snow samples were collected. $-1.10$ $-1.10$ $-1.10$ $-1.10$ $-1.10$ $-1.10$ $-1.10$ $0.01$ $-0.07$ $0.03$ $0.04$ $0.03$ $-0.08$                                                                                                                                                                                                                                                                                                                                                                                                                                                                                                                                                                                                                  | Comamonadaceae            | 0.06                       | 0.01            | 0.06           | 0.00           | -0.02       | 0.06            | 0.07              | 0.00              | 0.05           | -0.15               | 0.01          | 0.12    | -0.02               | -0.09     | 0.16      |
| Xanthomonadaceact $-0.10$ $0.01$ $0.07$ $0.07$ $0.09$ $-0.02$ $-0.03$ $-0.06$ $0.08$ $0.00$ $0.04$ $0.03$ $0.06$ Perudomonadaceact $0.18$ $0.00$ $0.02$ $-0.08$ $-0.10$ $0.17$ $0.01$ $-0.05$ $0.01$ $0.02$ $-0.08$ $-0.08$ $-0.08$ $0.04$ $0.03$ $0.04$ $0.03$ $0.04$ $0.03$ $0.04$ $0.03$ $0.04$ $0.03$ $0.04$ $0.06$ ** The marks $\uparrow$ indicates $P < 0.03$ $-0.07$ $0.03$ $0.04$ $0.03$ $0.04$ $0.03$ $0.04$ $0.04$ $0.04$ $0.04$ $0.04$ $0.04$ $0.04$ $0.04$ $0.04$ $0.04$ $0.04$ $0.04$ $0.04$ $0.04$ $0.04$ $0.04$ $0.04$ $0.04$ $0.04$ $0.04$ $0.04$ $0.04$ $0.04$ $0.04$ $0.04$ $0.04$ $0.04$ $0.04$ $0.04$ $0.04$ $0.04$ $0.04$ $0.04$ $0.04$ $0.04$ $0.04$ $0.04$ $0.04$ $0.04$ $0.04$ $0.04$ $0.04$ $0.04$ $0.04$ $0.04$ $0.04$ $0.04$ $0.04$ $0.04$ $0.04$ $0.04$ $0.04$ $0.04$ $0.04$ $0.04$ $0.04$ $0.04$ $0.04$ $0.04$ $0.04$ $0.04$ $0.04$ $0.04$ $0.04$ $0.04$ $0.04$ $0.04$ $0.04$ $0.04$ $0.04$ $0.04$ $0.04$ $0.04$ $0.04$ $0.04$ $0.04$ $0.04$ $0.04$ $0.04$ $0.04$ $0.04$ $0.04$                                                                                                                                                                                                                                                                                                                                                                                                                                                                                                                                               | Oxalobacteraceae          | -0.34††                    | -0.45††         | -0.51++        | -0.41++        | -0.42       | -0.02           | 0.33††            | -0.42++           | 0.56††         | -0.28               | -0.34††       | -0.26   | -0.33               | -0.24     | -0.87††   |
| Pseudomonadaceae     0.18     0.00     0.02     -0.08     -0.10     0.17     0.05     0.10     -0.07     0.03     0.04     0.03     -0.08     0.04       * The marks †* indicates P<0.01 and the mark † indicates P<0.05. Among the marked values, red cells indicate positive relations and blue cells indicate negative relations.                                                                                                                                                                                                                                                                                                                                                                                                                                                                                                                                                                                                                                                                                                                                                                                                                                                                                                                                                                                                                                                                                                                                                                                                                        | Xanthomonadaceae          | -0.10                      | 0.01            | 0.07           | 0.07           | 0.09        | -0.02           | 0.03              | -0.01             | -0.02          | -0.05               | 0.08          | 0.00    | 0.04                | 0.03      | 0.06      |
| * The marks †† indicates P<0.01 and the mark † indicates P<0.05. Among the marked values, red cells indicate positive relations and blue cells indicate negative relations.<br>** The temperature indicating the 50% of concentrations of ice-nucleic particles.<br>*** The heights (cm) of snow wall, from which snow samples were collected.                                                                                                                                                                                                                                                                                                                                                                                                                                                                                                                                                                                                                                                                                                                                                                                                                                                                                                                                                                                                                                                                                                                                                                                                              | Pseudomonadaceae          | 0.18                       | 0.00            | 0.02           | -0.08          | -0.10       | 0.17            | 0.01              | -0.05             | 0.10           | -0.07               | 0.03          | 0.04    | 0.03                | -0.08     | 0.04      |
| ** The temperature indicating the 50% of concentrations of ick-nucleic particles.                                                                                                                                                                                                                                                                                                                                                                                                                                                                                                                                                                                                                                                                                                                                                                                                                                                                                                                                                                                                                                                                                                                                                                                                                                                                                                                                                                                                                                                                           | * The marks †† indicate:  | 5 P<0.01 and               | the mark †      | indicates P<   | 0.05. Among t  | the marked  | values, red cel | lls indicate posi | tive relation     | s and blue c   | cells indicate      | negative rel: | ations. |                     |           |           |
| *** The heights (cm) of snow wall, from which snow samples were collected.                                                                                                                                                                                                                                                                                                                                                                                                                                                                                                                                                                                                                                                                                                                                                                                                                                                                                                                                                                                                                                                                                                                                                                                                                                                                                                                                                                                                                                                                                  | ** The temperature indi   | cating the 50 <sup>4</sup> | % of concen     | trations of ic | e-nucleic par  | ticles.     |                 |                   |                   |                |                     |               |         |                     |           |           |
|                                                                                                                                                                                                                                                                                                                                                                                                                                                                                                                                                                                                                                                                                                                                                                                                                                                                                                                                                                                                                                                                                                                                                                                                                                                                                                                                                                                                                                                                                                                                                             | *** The heights (cm) of s | mow wall, fre              | om which sn     | ow samples v   | vere collected |             |                 |                   |                   |                |                     |               |         |                     |           |           |

| 1 | 0 |  |
|---|---|--|
| I | 1 |  |