# Peer review of "Long-range transported bioaerosols captured in snow cover on Mount Tateyama, Japan: Impacts of Asian-dust events on airborne bacterial dynamics relating to ice-nucleation activities"

_Atmospheric Chemistry and Physics, 2017_

## Referee Comment (RC1) · Anonymous Referee #1 · 30 Jan 2018

A snow profile which had accumulated during winter and spring was sampled at 2450 m elevation on Mt. Tateyama, Japan. Seventy samples were analysed for ice nucleating particles, fluorescent particles, ion concentrations, and bacterial composition (16S rRNA). Samples from the lower part of the profile were characterised by particles mainly from northern parts of Asia and the Sea of Japan. Samples from the upper part contained larger numbers of particles that originated from the Asian deserts and industrial regions. I find the paper interesting and much of it is well done. At the same time, I would like to see more clarity in certain parts of it. Since I am not a biologist, I

can say little about the microbiological part of the study.

1.) Samples were filtered (0.22 micron) and re-suspended for INP quantification. I wonder what proportion of INP may have passed through the filter. Did you do, for comparison, drop freezing assays with samples prior to filtration?

2.) Equation at the end of page 4: What does "deltaMo" stand for? Does "V" stand for the volume of snow from which the particles were derived (e.g. is it corrected for the 200-fold concentration and dilution mentioned in the paragraph above the equation)?

3.) Page 5, line 3: Quantitative statements about ice nucleating particles (INP) should be accompanied by the temperature at which the mentioned INP were active. Otherwise these numbers have little meaning. The numbers (1.74 to 49.7 IN per litre) are very small compared to other numbers of INP in precipitation (please see for comparison the summarising Figure 1 in Petters and Wright, 2015, http://dx.doi.org/10.1002/2015GL065733). Is the unit (IN per litre) correct? Another unit that makes me wonder is the "m-3" on the y-axes of Fig. 4. Should this be "cm-3"?

4.) Page 9, lines 28-29: "Dust mineral particles without organic matters, such as ATD, showed lower temperatures (less than -15 °C) for the initial freezing of water drops than snow samples of the dirty layers." The onset of observed freezing or, as you call it: initial freezing, is a function of the particles' ice nucleation property and the total number of particles in the drop freezing assay. The same kind of particles will show a higher onset of observed freezing when a larger number (higher concentration) of them is tested in an assay (greater probability that it contains a rare INP active at warm temperature). Therefore, parameters like "initial freezing", "end-freezing" and "IN-T50C) are strictly relative numbers. They are meaningful when comparing samples that have all been processed exactly the same. In this context, it would be important to know exactly how the filtration, re-suspension and dilution (page 4, lines 21-24) was done and whether this procedure introduced differences between samples. You write: "Concentrated samples were diluted to the lowest particulate densities of approximately 5.0 ×
10ˆ4 particles mL-1 (from 1.0 $\mu$g mL−1 to 2.0 $\mu$g mL−1) using the nano-purewater, . . ." (page 4, lines 23-24). Does that mean you normalised samples to a particle density of 10ˆ4 mlˆ-1 for INP analysis?

5.) Similar to the previous comment about the onset of freezing depending on particle numbers in an assay, the ". . .higher diversity in the dirty snow layers than those of other snow layers. . ." (page 10, lines 8-9) could also result from a greater probability of identifying a rare species in a sample where a larger number of its copies are present (more dirty snow). What is the lowest number of copies of a species that would have been necessary for a species to be detected in your analysis?

6.) By looking at Figure 5a, I wonder why samples with high numbers of INP were not diluted and re-analysed to obtain INP values of all samples for at least one common reference temperature (e.g. -10 C).

Figure 4: Is the unit on the y-axes indeed "m-3"?

Table 1: Some headers and sentences are longer than the text boxes. What is "end-freezing temp."?

Table 2: Header: What do you mean with "Relatives of relative abundances . . ."? Foot-notes, **: What do you mean with ". . .the 50% of concentrations of ice nucleic particles."

The manuscript would benefit from English language editing.

---

## Referee Comment (RC2) · Anonymous Referee #2 · 13 Feb 2018

Comment on "Long-range transported bioaerosols captured in snow cover on Mount Tateyama, Japan: Impacts of Asian-dust events on airborne bacterial dynamics relating to ice-nucleation activities" by T. Maki et al.

It has been proven that bioaerosols have a significant impact on environment and climate over the past decades. In particular, bioaerosols could serve as active Ice Nuclei (IN), consequently affect the microphysical properties of cloud in the atmosphere. So, bioaerosol-cloud interaction in the atmosphere is known as important research topic for climate community. To investigate the ice-nucleation activities of bioaerosol and

[Figure]

Asian dust, this study presents aerosol chemical analysis and bioaerosols character­ization in snow at the altitudes of 2450m AGL over mount Tateyama in Japan. Then concentration and types of bioaerosols for dust events and non-dust events could be obtained from fluorescent microscopy and 16S rDNA sequencing analysis. The topic is of sufficient interest to the communities of study of atmospheric aerosol (especially bioaerosols) and climate change. In general, I find this manuscript to be of interest for publication and appropriate for ACP. There are several suggestions for improvement listed below that should be considered by the authors before publication.

1. Section 2: To make the readers more easily understand the method of your study, a flowchart that briefly summaries snow sampling and analysis processes is needed in the manuscript. Probably, 'Sampling and Methods' is better for the title of Section 2. Moreover, current title of section 2.2 is not appropriate because several analysis methods by use of ion chromatography, epifluorescence microscope as well as lidar are introduced.

2. Section 3 and 4: The authors are encouraged to combine these two sections to­gether. The current version is quite hard to get intact information of each subsection. Therefore, please rewrite and combine to a section.

3. Figure 3: lidar measurements at Toyama AD-net are used to show the periods of Asian dust events and air pollutions during February to April 2013. However, this figure cannot show sufficient information so that should be improved. The authors should use attenuated backscatter coefficient and depolarization ratio that could clearly show dust events and non-dust events, rather than retrieved extinction coefficient of spherical and non-spherical particles (soil dust).

4. Moreover, what is the altitude of lidar site at Toyama? According to position of red boxes in figure 3, it seems that the altitude of lidar site is very close to sea level. Now all dust events should be clearly seen based on lidar measurements, but it is not clear to distinguish local air pollution days from others. Please enlarge size of panels and

rescale the axis of figure 3.

5. Figures 4: there is a peak at 698-695cm of snow cover height for concentration of bacteria. Is it also a dust event? According to results from 16S rDNA sequencing analysis in figure 6, dust aerosols probably affected the sample. Please explain in the paper.

6. Line 3 in page 3: please change 'Taklamakan' to 'Taklimakan'.

7. Line 6 in page 3: please change 'Huang et al., 2015ab' to 'Huang J. et al., 2015; Huang Z. et al., 2015'.

8. Line 16 in page 5: change '36.57N, 137.60E' to '36.57°N, 137.60°E'.

9. Line 1 in page 6: I think 'coloured layers' is not suitable, 'polluted layers' and 'dirty layers' is much better. Please replace it throughout the manuscript.

10. Line 4 and 5 in page 7: depolarization ratio is more popular for lidar community than depolarization rates. Please change 'depolarization rates' to 'depolarization ratio' throughout the manuscript. Actually spherical-particle rates is included within lidar data in the paper, please rewrite it.

11. Line 5 in page 9: please change 'workers' to 'Researchers'.

12. To increase reader better understanding of impact of Asian dust and bioaerosols on climate over East Asia, please reference papers as follow.

Sugimoto, N., Z. Huang, T. Nishizawa, I. Matsui, and B. Tatarov, 2012: Fluorescence from atmospheric aerosols observed with a multi-channel lidar spectrometer, Optics Express, 20(19), 20800-20807.

Huang J., Y. Li, C. Fu, F. Chen, Q. Fu, A. Dai, M. Shinoda, Z. Ma, W. Guo, Z. Li, L. Zhang, Y. Liu, H. Yu, Y. He, Y. Xie, X. Guan , M. Ji, L. Lin, S. Wang, H. Yan and G. Wang, 2017: Dryland climate change recent progress and challenges. Reviews of Geophysics, 55, 719-778, doi:10.1002/2016RG000550.

[Figure]

Huang J., H. Yu , A. Dai, Y. Wei, and L. Kang, 2017: Drylands face potential threat under 2°C global warming target. Nature Climate Change, doi: 10.1038/NCLIMATE3275.

Tang, K., Huang, Z., Huang, J., Maki, T., Zhang, S., Ma, X., Shi, J., Bi, J., Zhou, T., Wang, G., and Zhang, L.: Characterization of atmospheric bioaerosols along the transport pathway of Asian dust during the Dust-Bioaerosol 2016 Campaign, Atmos. Chem. Phys. Discuss., https://doi.org/10.5194/acp-2017-1172, in review, 2017.

13. The results in the paper give us further information about bioaerosols in snow, especially affected by Asian dust events. The authors are encouraged to evaluate the impact of bioaerosols on surface albedo and melting rate of snow in future.

Please also note the supplement to this comment:
https://www.atmos-chem-phys-discuss.net/acp-2017-1241/acp-2017-1241-RC2-supplement.pdf

―――――――――――――――

---

## Author Comment (AC1) · 15 Apr 2018

Dear Anonymous Referee 1,

I appreciate your kindly and useful comments for our manuscript. Moreover, we are very glad that our study has been valued. Furthermore, I am sorry for bothering you due to some mistake in the description of ice-nuclei experimental design. I would have revised our manuscript referring to your comments, and wish your review again. Your comments are indicated at sections (Q) and my responses are indicated at sections

(A). In sections (A), the revised parts in our manuscript were indicated using line.

Q1.) Samples were filtered (0.22 micron) and re-suspended for INP quantification. I wonder what proportion of INP may have passed through the filter. Did you do, for comparison, drop freezing assays with samples prior to filtration?

A1: After the particulate matters were removed from some of snow samples using 0.22 micron filters, the samples without >0.22 micron particulate matters showed mostly similar IN activities to nano-puresawater. Moreover, I have compared the melted snow samples without filtration and with re-suspension. There is no significant difference between them. Accordingly, I think that the soluble substrates in snow samples can be neglected in this study. I have added this explanation in the revised manuscript (P5 L6-L9).

Q2.) Equation at the end of page 4: What does "deltaMo" stand for? Does "V" stand for the volume of snow from which the particles were derived (e.g. is it corrected for the 200-fold concentration and dilution mentioned in the paragraph above the equation)?

A2: Sorry for occurring this confusion, because the explanation about the equation are insufficient. The factor "deltaMo" had meant the dilution rate in the previous version. I have revised the equation and inserted the dilution and concentration factor "C" instead of "deltaMo". The explanation about "C" has been added (P5 L5).

Q3.) Page 5, line 3: Quantitative statements about ice nucleating particles (INP) should be accompanied by the temperature at which the mentioned INP were active. Otherwise these numbers have little meaning. The numbers (1.74 to 49.7 IN per litre) are very small compared to other numbers of INP in precipitation (please see for comparison the summarising Figure 1 in Petters and Wright, 2015, http://dx.doi.org/10.1002/2015GL065733). Is the unit (IN per litre) correct? Another unit that makes me wonder is the "m-3" on the y-axes of Fig. 4. Should this be "cm-3"?

A3: I am sorry again for confusing you. The INP numbers have to be shown using "mL-
1". I have integrated the unit of INP concentration to "L-1" in the revised manuscript (P5 L6). Moreover, the previous Figure has not real INP numbers. I have calculated them again and inserted additional figure (Figures 5 and 7). I would like to appreciate your comments.

Q4.) Page 9, lines 28-29: "Dust mineral particles without organic matters, such as ATD, showed lower temperatures (less than -15 C) for the initial freezing of water drops than snow samples of the dirty layers." The onset of observed freezing or, as you call it: initial freezing, is a function of the particles' ice nucleation property and the total number of particles in the drop freezing assay. The same kind of particles will show a higher onset of observed freezing when a larger number (higher concentration) of them is tested in an assay (greater probability that it contains a rare INP active at warm temperature). Therefore, parameters like "initial freezing", "end-freezing" and "IN-T50C) are strictly relative numbers. They are meaningful when comparing samples that have all been processed exactly the same. In this context, it would be important to know exactly how the filtration, re-suspension and dilution (page 4, lines 21-24) was done and whether this procedure introduced differences between samples. You write: "Concentrated samples were diluted to the lowest particulate densities of approximately 5.0 10ËE4 particles mL-1 (from 1.0 g mL-1 to 2.0 g mL-1) using the nano-purewater, : : :" (page 4, lines 23-24). Does that mean you normalised samples to a particle density of 10ËE4 mlËE-1 for INP analysis?

A4: Thank you for your indication. We adjusted the particles concentrations in snow samples to same in dependence on DAPI-count densities. We would like to compare the IN activities under same particles concentrations 5.0 10ËĘ4 particles mL-1. The parameters such as "initial freezing", "end-freezing" and "IN-T50C) are show the relative abilities of IN in snow samples. The dilution and concentration procedures have been described in detail in the revised manuscript (P4 L24-L29)

Q5.) Similar to the previous comment about the onset of freezing depending on particle numbers in an assay, the ": : :higher diversity in the dirty snow layers than those of
other snow layers: : :" (page 10, lines 8-9) could also result from a greater probability of identifying a rare species in a sample where a larger number of its copies are present (more dirty snow). What is the lowest number of copies of a species that would have been necessary for a species to be detected in your analysis?

A5: I think MiSeq sequencing provided enough read numbers that almost bacterial categories (species) can be followed in this study. In fact, rarefaction curve showed that the bacterial OUT numbers are saturated at the numbers of analyzed sequences. In general, the lowest number of sequences in minor category (species) are single sequence and the almost saturation of OUT numbers is indicator for judging the follow of entire bacterial categories. I have added this explanation about the minor categories of OUT (P9 L10-L12).

Q6.) By looking at Figure 5a, I wonder why samples with high numbers of INP were not diluted and re-analysed to obtain INP values of all samples for at least one common reference temperature (e.g. -10 C).

A6: I have shown the IN numbers using other figure. Sorry. The Y axis did not show IN numbers and indicate the freezing well numbers (Figure). This graph is needed for the determination of IN-T50C and should be remained after revision (Figure 6a).

Q7. Figure 4: Is the unit on the y-axes indeed "m-3"?

A7: The Y-axes indicated the particle concentrations in melted snow samples (liquid). Accordingly, the unit is "mL-1". I have shown the concentrations using the unit "L-1" in the figures (Figures 5 and 7).

Q8. Table 1: Some headers and sentences are longer than the text boxes. What is "endfreezing temp."?

A8: Some headers and sentences have revised and all parts can be shown in the table in the revised manuscript (Table). The term "endfreezing temp." has been explained in this table (Table 1).

**ACPD**
Q9. Table 2: Header: What do you mean with "Relatives of relative abundances : : :"? Footnotes, \*\*: What do you mean with ": : :the 50% of concentrations of ice nucleic particles."

A8: I have revised the header. Moreover, some explanations in table have been redrafted (Table 2).

Q10. The manuscript would benefit from English language editing.

A8: English language has been checked by native speakers again (Entire section of the revised manuscript).
**Black carbon** Concentrations of bacteria (copies L-1) Blue Yellow ≥5 µm Yellow <5 µm 108 White ≥5 µm White <5 µm 107

The heights of snow wall (cm)

Densities of particles (x109 particles L-1)

Δ

3

Fig. 5 T. Maki et al.

109

Fig. 1. Figure 5: Vertical profiles for DAPI-stained particle densities (bars) and 16S rRNA genes copies determined by qRT-PCR (open circles), in snow samples collected from Murododaira, Mt. Tateyama, in Apri
**ACPD**
Fig. 7 T. Maki et al.

**Fig. 2.** Figure 7: Variations of ice-nuclei particles in the snow samples collected from the dirty layers (orange lines) and non-dirty layer (grey lines) in the upper parts and all the layers in lower parts (g

---

## Author Comment (AC2) · 15 Apr 2018

Dear Anonymous Referee 2,

I appreciate your useful comments and valuable suggestions for our manuscript. Moreover, we feel very glad that our study has been valued. I would have revised our manuscript referring to your comments, and wish your review again. Your comments are indicated at sections (Q) and my responses are indicated at sections (A). In sections (A), the revised parts in our manuscript were indicated using line.

Q1. Section 2: To make the readers more easily understand the method of your study, a flowchart that briefly summaries snow sampling and analysis processes is needed in the manuscript. Probably, 'Sampling and Methods' is better for the title of Section 2. Moreover, current title of section 2.2 is not appropriate because several analysis methods by use of ion chromatography, epifluorescence microscope as well as lidar are introduced.

A1: As your suggestions, I have revised the titles in Section 2 (P3 L20, P4 L3) and inserted the flowchart figure (Figure 2).

Q2. Section 3 and 4: The authors are encouraged to combine these two sections together. The current version is quite hard to get intact information of each subsection. Therefore, please rewrite and combine to a section.

A2: I agree with your comment. Section 3 and 4 have been combined in the revised manuscript (Section of Results and Discussion).

Q3. Figure 3: lidar measurements at Toyama AD-net are used to show the periods of Asian dust events and air pollutions during February to April 2013. However, this figure cannot show sufficient information so that should be improved. The authors should use attenuated backscatter coefficient and depolarization ratio that could clearly show dust events and non-dust events, rather than retrieved extinction coefficient of spherical and non-spherical particles (soil dust).

A3: The data of attenuated backscatter coefficient and depolarization ratio are used in the lidar figure in the revised manuscript (Figures 4 and S1).

Q4. Moreover, what is the altitude of lidar site at Toyama? According to position of red boxes in figure 3, it seems that the altitude of lidar site is very close to sea level. Now all dust events should be clearly seen based on lidar measurements, but it is not clear to distinguish local air pollution days from others. Please enlarge size of panels and rescale the axis of figure 3.
A4: The lidar site of Toyama is located at the altitude of 25 m at sea level. This altitude can be neglected in the lidar figure (Figure 4).

Q5. Figures 4: there is a peak at 698-695cm of snow cover height for concentration of bacteria. Is it also a dust event? According to results from 16S rDNA sequencing analysis in figure 6, dust aerosols probably affected the sample. Please explain in the paper.

A5: This layer indicated high concentrations of total particles as well as bacterial cells. I think this layers would include some particles transported by very short term of dust events, which had not detected using lidar measurements. This suggestion has been added in the revised manuscript (P7 L27-L29).

Q6. Line 3 in page 3: please change 'Taklamakan' to 'Taklimakan'.

A6: I have integrated to 'Taklimakan' (Entire section of the revised manuscript).

Q7. Line 6 in page 3: please change 'Huang et al., 2015ab' to 'Huang J. et al., 2015; Huang Z. et al., 2015'.

A7: Thank you for your advice. I have changed to this citation style (P2 L5, P7 L1).

Q8. Line 16 in page 5: change '36.57N, 137.60E' to '36.57N, 137.60E'.

A8: As your comment, I have revised the description of longitude and latitude (P3 L22).

Q9. Line 1 in page 6: I think 'coloured layers' is not suitable, 'polluted layers' and 'dirty layers' is much better. Please replace it throughout the manuscript.

A9: As your suggestion, I have integrated to use the term 'dirty layers' at entire sections (Entire section of the revised manuscript).

Q10. Line 4 and 5 in page 7: depolarization ratio is more popular for lidar community than depolarization rates. Please change 'depolarization rates' to 'depolarization ratio' throughout the manuscript. Actually spherical-particle rates is included within lidar data

**ACPD**
in the paper, please rewrite it.

A10: Thank you for your suggestion. I have integrated to use the term 'depolarization ratio' (Entire section of the revised manuscript).

Q11. Line 5 in page 9: please change 'workers' to 'Researchers'.

A11: I have changed 'workers' to 'Researchers' (P5 L30).

Q12. To increase reader better understanding of impact of Asian dust and bioaerosols on climate over East Asia, please reference papers as follow. Sugimoto, N., Z. Huang, T. Nishizawa, I. Matsui, and B. Tatarov, 2012: Fluorescence from atmospheric aerosols observed with a multi-channel lidar spectrometer, Optics Express, 20(19), 20800-20807. Huang J., Y. Li, C. Fu, F. Chen, Q. Fu, A. Dai, M. Shinoda, Z. Ma, W. Guo, Z. Li, L. Zhang, Y. Liu, H. Yu, Y. He, Y. Xie, X. Guan , M. Ji, L. Lin, S. Wang, H. Yan and G. Wang, 2017: Dryland climate change recent progress and Huang J., H. Yu , A. Dai, Y. Wei, and L. Kang, 2017: Drylands face potential threat under 2C global warming target. Nature Climate Change, doi: 10.1038/NCLIMATE3275. Tang, K., Huang, Z., Huang, J., Maki, T., Zhang, S., Ma, X., Shi, J., Bi, J., Zhou, T., Wang, G., and Zhang, L.: Characterization of atmospheric bioaerosols along the transport pathway of Asian dust during the Dust-Bioaerosol 2016 Campaign, Atmos. Chem. Phys. Discuss., https://doi.org/10.5194/acp-2017-1172, in review, 2017.

A12: Thank you for telling us important reference papers. I referred these references in the revised manuscript (Entire section of the revised manuscript).

Q13. The results in the paper give us further information about bioaerosols in snow, especially affected by Asian dust events. The authors are encouraged to evaluate the impact of bioaerosols on surface albedo and melting rate of snow in future.

A12: I appreciate your valuable comments for the perspectives to our research. This comments were also described in the conclusion section of the revised manuscript (P11 L21-22).

**ACPD**
Fig. 1. Figure2: Sampling and experimental scheme.